# A core phyllosphere microbiome exists across distant populations of a tree species indigenous to New Zealand

**Anya S. Noble, Stevie Noe, Michael J. Clearwater, Charles K. Lee** *

School of Science, University of Waikato, Hamilton, New Zealand

* charles.lee@waikato.ac.nz

## Abstract

The phyllosphere microbiome is increasingly recognised as an influential component of plant physiology, yet it remains unclear whether stable host-microbe associations generally exist in the phyllosphere. *Leptospermum scoparium* (mānuka) is a tea tree indigenous to New Zealand, and honey derived from mānuka is widely known to possess unique antimicrobial properties. However, the host physiological traits associated with these antimicrobial properties vary widely, and the specific cause of such variation has eluded scientists despite decades of research. Notably, the mānuka phyllosphere microbiome remains uncharacterised, and its potential role in mediating host physiology has not been considered. Working within the prevailing core microbiome conceptual framework, we hypothesise that the phyllosphere microbiome of mānuka exhibits specific host association patterns congruent with those of a microbial community under host selective pressure (null hypothesis: the mānuka phyllosphere microbiome is recruited stochastically from the surrounding environment). To examine our hypothesis, we characterised the phyllosphere and associated soil microbiomes of five distinct and geographically distant mānuka populations across the North Island of New Zealand. We identified a habitat-specific and relatively abundant core microbiome in the mānuka phyllosphere, which was persistent across all samples. In contrast, non-core phyllosphere microorganisms exhibited significant variation across individual host trees and populations that was strongly driven by environmental and spatial factors. Our results demonstrate the existence of a dominant and ubiquitous core microbiome in the phyllosphere of mānuka, supporting our hypothesis that phyllosphere microorganisms of mānuka exhibit specific host association and potentially mediate physiological traits of this nationally and culturally treasured indigenous plant. In addition, our results illustrate biogeographical patterns in mānuka phyllosphere microbiomes and offer insight into factors contributing to phyllosphere microbiome assembly.

## Introduction

Plants harbour distinct and dynamic microhabitats colonised by complex microbial communities known as plant *microbiomes* [1]. Evidence has shown that interactions between plants and

**Data Availability Statement:** DNA sequences generated for the current study are available from the NCBI Sequence Read Archive (SRA) under the accession number PRJNA556542.

**Funding:** ASN was supported by the University of Waikato Doctoral Scholarship.

**Competing interests:** The authors have declared that no competing interests exist.

their associated microorganisms in many of these microbiomes play a pivotal role in host plant health, function, and evolution [2]. The leaf surface, or phyllosphere, harbours a microbiome comprising diverse communities of bacteria, fungi, algae, archaea, and viruses [3, 4]. Interactions between the host plant and phyllosphere bacteria have the potential to drive various aspects of host plant physiology [5–7]. However, our knowledge of these bacterial associations in the phyllosphere remains relatively modest, and there is a need to advance fundamental knowledge of phyllosphere microbiome dynamics [8].

The assembly of the phyllosphere microbiome, herein strictly defined as epiphytic bacterial communities on the leaf surface, can be shaped by the microbial communities present in the surrounding environment (i.e., stochastic colonisation) and the host plant (i.e., biotic selection) [3, 9]. However, although the leaf surface is generally considered a discrete microbial habitat [10, 11], there is no consensus on the dominant driver of community assembly across phyllosphere microbiomes. For example, host-specific bacterial communities have been reported in the phyllosphere of co-occurring plant species, suggesting a dominant role of host selection [11–13]. Conversely, microbiomes of the surrounding environment have also been reported to be the primary determinant of phyllosphere community composition [10, 14–16]. As a result, the processes that drive phyllosphere community assembly are not well understood but unlikely to be universal across plant species. However, the existing evidence does indicate that phyllosphere microbiomes exhibiting host-specific associations are more likely to interact with the host than those primarily recruited from the surrounding environment [5, 17–19].

The search for a core microbiome in host-associated microbial communities is a useful first step in trying to understand the interactions that may be occurring between a host and its microbiome [20, 21]. The prevailing core microbiome concept is built on the notion that the persistence of a taxon across the spatiotemporal boundaries of an ecological niche is directly reflective of its functional importance within the niche it occupies; it therefore provides a framework for identifying functionally critical microorganisms that consistently associate with a host species [20, 22, 23]. However, divergent definitions of "core microbiome" have arisen across scientific literature with researchers variably identifying "core taxa" as those persistent across distinct host microhabitats [24, 25] and even different species [13, 17]. Here we assert that, given the functional divergence of microorganisms across different host species [13] and microhabitats [26], defining core taxa *sensu stricto* as those persistent across broad geographic distances within tissue- and species-specific host microbiomes, represents the most biologically and ecologically appropriate application of this conceptual framework [27]. To our knowledge, tissue- and species-specific core microbiomes across host populations separated by broad geographical distances have not been widely reported for the phyllosphere using the stringent definition established by Ruinen (4).

*Leptospermum scoparium* var. *scoparium* (Myrtaceae), commonly known as "*mānuka*", is a flowering tea tree indigenous to New Zealand [28]. Mānuka honey, produced from the nectar of mānuka flowers, is globally renowned for its unique non-peroxide antibacterial properties [29, 30]. These non-peroxide antibacterial properties have been principally linked to the accumulation of the three-carbon sugar dihydroxyacetone (DHA) in the nectar of the mānuka flower, which undergoes a chemical conversion to methylglyoxal (MGO) in mature honey [31–33]. However, the concentration of DHA in the nectar of mānuka flowers is notoriously variable, and the antimicrobial efficacy of mānuka honey consequently varies from region to region and from year to year [34–36]. Despite extensive research efforts, no reliable correlation has been identified between DHA production and climatic [37], edaphic [38], or host genetic factors [39]. Microorganisms have been studied in the mānuka rhizosphere and endosphere [40–42]. Although previous studies have primarily focussed on fungi, a recent study provided the first investigation of endophytic bacterial communities from three geographically and

environmentally distinct mānuka populations using fingerprinting techniques and revealed tissue-specific core endomicrobiomes [43]. However, a similar characterisation of the mānuka phyllosphere microbiome has not been conducted.

Inspired by the intriguing physiological characteristics of the mānuka tree and the increasingly recognised role of phyllosphere bacteria in plant physiology, we provide the first characterisation of the bacterial communities comprising the mānuka phyllosphere microbiome. Working within the prevailing core microbiome conceptual framework, we hypothesise that phyllosphere microorganisms of mānuka exhibit specific host association patterns congruent with those of a microbial community under host selective pressure. Correspondingly, our null hypothesis is that the mānuka phyllosphere microbiome is recruited stochastically from the surrounding environment. To test this hypothesis, we characterised the phyllosphere and associated soil bacterial communities of five distinct and geographically distant mānuka populations across the North Island of New Zealand using 16S rRNA gene PCR amplicon sequencing. Our findings revealed a dominant and ubiquitous core microbiome, providing strong evidence for a specific host association. This knowledge may be useful for designing future studies that will enhance our understanding of plant-microbe associations in the phyllosphere and address significant and longstanding questions such as the factors driving spatiotemporal variability in mānuka DHA production.

## Material and methods

### Selection of *Leptospermum scoparium* (mānuka) populations

Five indigenous mānuka populations across the North Island of New Zealand were selected for study: Mohaka, Hawkes Bay (39˚01 S; 177˚08 E), Serpentine Lake, Waikato (37˚56 S, 175˚19 E), Mangatarere Valley, Wellington (40˚57 S; 175˚26 E), Mamaku, Bay of Plenty (38˚02 S; 176˚03 E), and North-eastern Kaimanawas, Taupo (39˚06 S; 176˚21 E) (S1 Fig). Steens Honey provided access to the sites in Mohaka and Mangatarere Valley. The East Taupō Lands Trust provided access to the north-eastern Kaimanawa site, Timberlands Ltd provided access to the Mamaku site. The Waipa District Council provided access to the site at Serpentine Lake. Straight-line distances between sites range from 65 km (Serpentine Lake and Mamaku) to 333 km (Serpentine Lake and Mangatarere Valley). Monthly climate data for each site was retrieved from the National Climate Database (NIWA) (https://cliflo.niwa.co.nz/).

### Sample collection

Each site was sampled once during the November 2016 –January 2017 mānuka flowering season. Sampling times were staggered in correspondence with flower opening in each region [37]. Per site, three branches with approximately 100–200 leaves were chosen from each of six seemingly healthy mānuka trees that displayed no visible signs of disease or damage. Each branch was cut with clippers sterilised on site using 70% v/v ethanol/water, placed in an individual sterile zip lock bag, and immediately put on ice. Surface soil (1–2 mm) from multiple positions around the base of each sample tree was also collected into sterile 50 mL Falcon tubes using a spatula sterilised on site using 70% v/v ethanol/water and immediately placed on ice. Upon return to the Thermophile Research Unit at the University of Waikato, branch and soil samples were frozen at -20˚C and -80˚C, respectively, until further analysis.

### Environmental and host tree metadata

At each site, a datalogger (CR10X, Campbell Scientific, Utah) and sensors was used to record air temperature, relative humidity (Humitter 50Y, Vaisala, Finland), and photosynthetically

active radiation (PPFD) (LI-190, Licor, Utah) every 15 minutes over 24 h prior to sample collection. Metadata was also collected for each sample tree, these included: GPS coordinates (WGS84 (G1762); degree minutes), elevation, tree height, and the basal diameter of the tree base at 10 cm off the ground. For each of the three branches collected per tree, height of the branch above ground and aspect (0–360˚) were also recorded.

## Bacterial community collection

Per branch, 1 g of predominantly healthy, undamaged green leaves were aseptically excised and pooled. Leaves were submerged in 10 mL of wash buffer (phosphate buffer solution [PBS, 100mM NaH2PO4], 1% tween 20) and sonicated for 20 min (60 Hz) in an ultrasonic cleaning bath. Sonication was used to remove bacteria from the leaf surface in order to minimise undesirable amplification of chloroplast rRNA genes and contamination by endophytic bacteria, which are typically recovered via methods that inflict damage to leaves such as maceration and coring [44, 45]. After sonication, the wash buffer was decanted, syringe filtered through 90 μm nylon mesh to remove finer plant debris, and centrifuged (3,200 x g for 30 minutes). The supernatant was discarded, and the bacterial cell pellet was resuspended in 270 μl of PBS, transferred to a 1.5–2.0 mL screw-capped conical bottomed polypropylene tube containing 0.5 g each of 0.1 mm and 2.5 mm silica-zirconia beads, and frozen at -80˚C until further processing. Total genomic DNA was extracted from the microbial cell suspension of each leaf sample using a modified version of a cetyl trimethylammonium bromide (CTAB) bead beating protocol, which has been shown to be highly effective for low biomass samples [46]. The Power Soil DNA Extraction kit (MoBio) was used to extract DNA from 0.5 g of each soil sample collected from the base of each mānuka tree. Extracted DNA was quantified using the QuBit-IT dsDNA HS Assay Kit (Life Technologies, Auckland) and stored at -20˚C until further analysis.

## DNA library preparation and sequencing

Total community DNA extracted from the phyllosphere and surface soil was used for amplification and sequencing of the V4 region of the 16 rRNA gene using the universal primer set F515 (5′GTGCCAGCMGCCGCGGTAA–3′) and R926 (5′–CCACTACGCCTCCGCTTTCCTC TCTATGGGCAGTCGGTGATCCGYCAATTYMTTTRAGTTT–3′). Sample cross-contamination via aerosolised PCR products is a major limitation of two-step protocols widely used in preparing 16S rRNA gene PCR amplicons for Illumina MiSeq sequencing [47–51]. To avoid cross-contamination of aerosolised PCR products, we used fusion primers to prepare 16S rRNA gene PCR amplicons in a one-step protocol. Unlike previous studies, chloroplast-excluding primers were avoided in order to prevent taxonomic bias and permit identification of Cyanobacteria in the mānuka phyllosphere [52–54]. The fusion primers used in the current study generated an amplicon of approximately 500 bps with adapters suited for Ion Torrent sequencing. PCR amplification was performed in 20-μl reactions consisting of 0.8 μL bovine serum albumin (BSA) (Promega Corporation, USA), 2.4 μL dNTPs (2mM each) (Invitrogen Ltd, New Zealand), 2.4 μL 10x PCR buffer (Invitrogen), 2.4 μL MgCl2 (50 mM) (Invitrogen), 0.4 μL each primer (10 mM) (Integrated DNA Technologies, Inc), 0.096 μL Taq DNA polymerase (Invitrogen), 2 μL of genomic DNA (2.5 ng/μl), and 9.104 μL molecular-grade water. Reactions were performed in triplicate for each sample with the following thermocycler conditions: 3-min initial denaturation at 94˚C, followed by 30 cycles of 45 s at 94˚C, 1-min at 50˚C, and 1.5-min at 72˚C, with a final 10-min elongation at 72˚C. For each run, a positive and a negative control was included. The PCR products were cleaned and normalized using a SequelPrep Normalization Kit, (Life Technologies, Auckland) according to the manufacturer's instructions.

DNA sequencing was undertaken at the Waikato DNA Sequencing Facility at the University of Waikato using an Ion Torrent PGM DNA sequencer with an Ion 318v2 chip (Life Technologies). Raw sequences in FASTQ format were first filtered in Mothur to remove short reads, long reads, and reads with excessive homopolymers [55]. Thereafter, sequences were quality filtered using USEARCH (ver 9) [56]. A total of 1,890,959 high quality reads were obtained and clustered into OTUs at a percent sequence similarity threshold of 97%. After filtering for chloroplast OTUs, a total 928,317 reads and an average 10,430 (mean) reads per mānuka sample remained (n = 89). These reads mapped to 1384 bacterial OTUs (97%). Meanwhile, a total 736,849 reads and an average 25,409 (mean) reads per soil sample remained (n = 29). These reads mapped to 6905 bacterial OTUs (97%). From this initial processing, a BIOM file, FASTA file and OTU table were generated. The raw FASTA file was run through the Michigan State University Ribosomal Database Project (RDP) Classifier whereby taxonomy was assigned to the 16S rRNA sequences [57]. Taxonomic assignments with estimated confidence less than 80% were classified as 'unknown'. Taxonomy data was merged with the BIOM file using the 'biom add-metadata command' in The BIOM file format (ver 2.1). A rarefied dataset was generated by rarefying reads on a per-sample basis to the 5th percentile of per-sample sequence reads (3,979) using rarefy_even_depth implemented in the phyloseq package [58]. Five samples were excluded from rarefaction due to low numbers of sequence reads.

## Statistics and data analysis

Data analyses and visualisation were performed in R version 3.4.3 [59], with the packages DESeq2 [60], ggplot2 [61], phyloseq [58], and vegan [62]. Observed OTU richness in combination with the Shannon diversity index and the Chao1 index were used as measures of alpha diversity, and beta diversity was computed using the Bray-Curtis dissimilarity index [63]. The Euclidean index was used to compute distances between spatial and environmental parameters [64]. Differences in alpha diversity between regions were compared using analysis of variance (ANOVA) and correlations between alpha diversity and explanatory variables was analysed using the Pearson correlation coefficient. A significance level of 0.05 was used in all analyses. The beta diversity between different sites, trees, and sample type was compared using permutational multivariate analysis of variance (PERMANOVA) (Adonis from the package vegan with 999 permutations) [62, 65]. Partial Mantel tests using the Pearson product-moment correlation coefficient were used to test the correlation between total, core, and non-core community (Bray-Curtis) and environmental (Euclidean) distance matrices [66]. Canonical redundancy analysis (RDA) and variance partitioning were used to describe and partition variation in bacterial community structure among three groups of explanatory variables: 'Spatial', 'Environmental', and 'Host'. The group of environmental variables included the night-day temperature differential, night temperature, monthly precipitation, and monthly cloud cover. The spatial explanatory variable corresponded to the straight-line distance between sample coordinates (UTM). Host variables included tree height and tree diameter. These variables were chosen as they represented the largest correlations with community composition and diversity in Partial Mantel analysis. The nucleotide Basic Local Alignment Search Tool (BLASTn) algorithm was used to identify isolates with the closest sequence identity to core taxa. Core and BLASTn sequences with the highest nucleotide identity (%) were aligned with ClustalW. A phylogenetic tree was constructed in MEGA X [67] using Maximum Likelihood method and Tamura-Nei model [68]. The R package 'DESeq2' was used to identify OTUs that are differentially abundant across phyllosphere and soil communities and obtain estimates of log-fold changes [60].

### The core microbiome

The core mānuka phyllosphere microbiome, was identified by plotting OTU abundance and occupancy. An OTU presence in all 89 phyllosphere samples (100% occupancy) was chosen as a highly conservative representation of the core microbiome. In selecting 100% occupancy as the stringent core microbiome criteria, we minimised the likelihood of these taxa representing transient taxa [69]. The core phyllosphere microbiome at 100% occupancy was annotated in both rarefied and non-rarefied datasets.

## Results

### Sample sites

Leaf and soil samples were collected from five native mānuka populations across the North Island of New Zealand (S1 Fig). These sampling sites were geographically distinct, separated by linear distances ranging from 65 to 333 km, and situated at elevations of 66 to 634 m above sea level (ASL). Measurements of air temperature, relative humidity, and photosynthetically active radiation fluctuated across the 24 hours preceding sampling and were variable between populations (S2 Fig and S1 Table). Across all sampling sites, 29 healthy *Leptospermum scoparium* trees of differing heights (1.8 to 4.1 m) and basal diameters (1.5 to 24.8 cm) were selected. In total, 89 mānuka branches were collected from the lower canopy (0.6 to 2.8m) (S2 Table).

### Structure of the mānuka phyllosphere microbiome

Sequencing of bacterial 16S rRNA gene PCR amplicons yielded 10,430±5,165 reads per branch after removing chloroplast reads (on average 11.9% of the quality-assured reads). Across all samples, 1,384 operational taxonomic units (OTUs) were identified at an average of 256±63 OTUs per branch (S3 Table). Alpha diversity was calculated using richness, Shannon, and Chao1 indices (S4 Table). Phyllosphere communities of different sites exhibited statistically significant differences in richness (ANOVA, $P = 2.99 \times 10^{-6}$, F = 9.264) and Chao1 (ANOVA, P = 0.00231, F = 4.532) (S3 Fig and S5 Table). Alpha-diversity indices exhibited a significant and positive linear relationship with variables indicative of tree age such as tree height and diameter (S6 Table). The spatial characteristics of sampled branches relative to the trees, such as branch height and aspect, exhibited weak but significant correlations with observed OTU richness and Chao1 (S6 Table). Meanwhile, monthly average temperature and monthly sun hours were the only environmental variables to exhibit a significant correlation with alpha diversity, both demonstrating negative and weak relationships (S6 Table).

In total, 21 phyla were identified in the collective mānuka phyllosphere. Taxa belonging to the phyla Proteobacteria, Acidobacteria, Bacteriodetes, Firmicutes, and Verrucomicrobia consistently represented the largest proportion of reads in all samples, of which Proteobacteria consistently showed the greatest relative abundance (S4 Fig). Within Proteobacteria, Alphaproteobacteria was the most dominant, representing 46.5% of all reads, followed by Gammaproteobacteria (3.2%), Betaproteobacteria (0.75%) and Deltaproteobacteria (0.34%).

### Mānuka phyllosphere core microbiome

Of the 1,384 bacterial OTUs identified in the total phyllosphere community, 10 OTUs were identified in all phyllosphere samples (100% occupancy) and therefore defined as members of the core phyllosphere microbiome (Fig 1D). These 10 core taxa were affiliated with four phyla and five classes: Alphaproteobacteria (Rhizobiales [29.3% of all reads], Sphingomonadales [4.9%]), Bacteroidetes (Cytophagia [2.5%]), Verrucomicrobia (Spartobacteria [2.4%]), and Acidobacteria (Acidobacteriia [1.0%]) (Fig 1A–1C). All 10 core OTUs were relatively

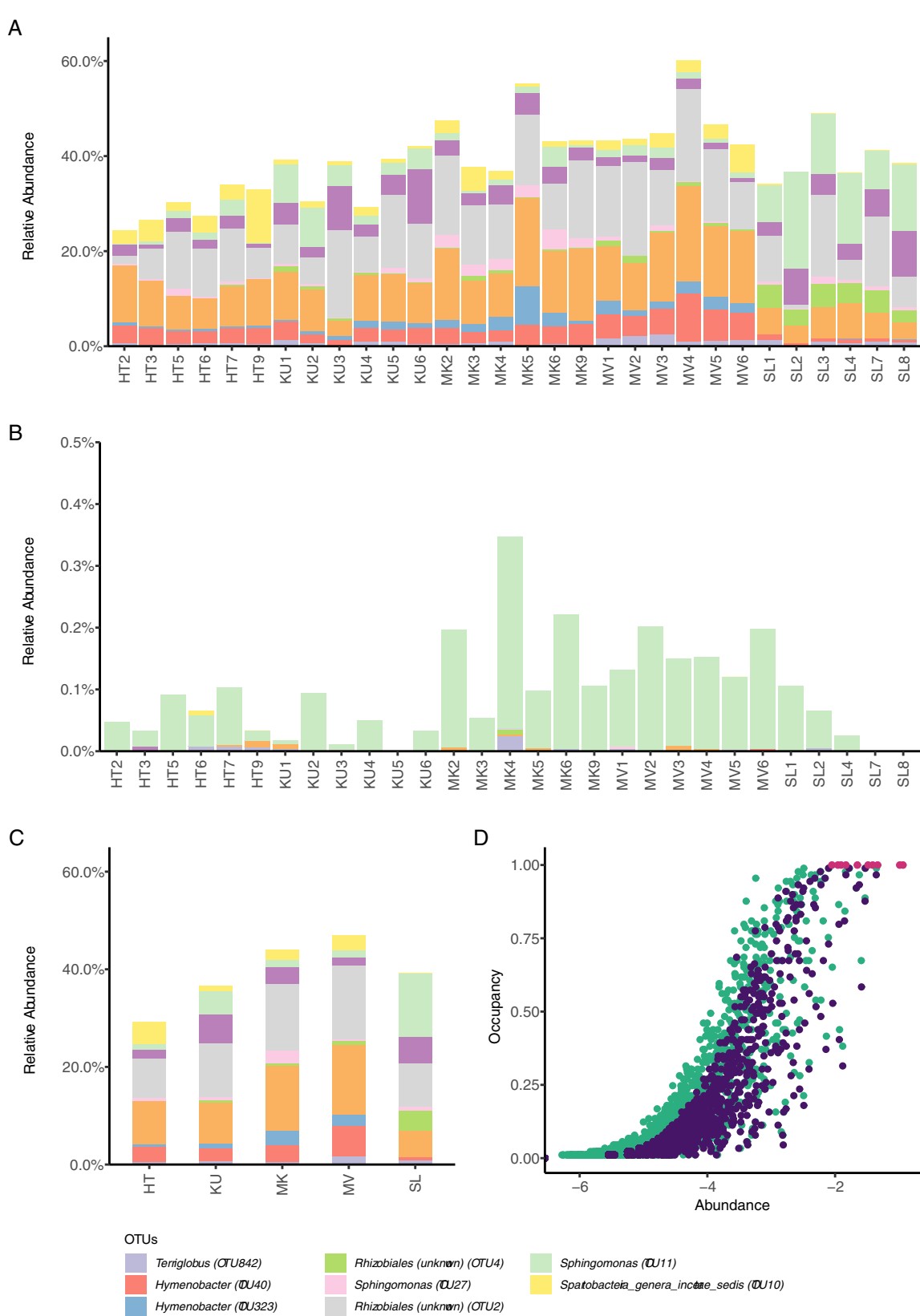

**Fig 1.** Relative abundance of core phyllosphere taxa in the mānuka phyllosphere (A, C) and associated soil communities (B). Relative abundance is averaged per tree (A) and site (C). An abundance-occupancy distribution was used to identify core phyllosphere taxa in non-rarefied (green) and rarefied (purple) datasets (D). Each point represents a taxon plotted by its mean log10 relative abundance and occupancy. Taxa (pink) with an occupancy of 1 (i.e., detected in all 89 phyllosphere samples) were considered members of the core microbiome. Panels A–C are colour-coded by OTU.

abundant and collectively represented 40.7% of all reads (S7 Table). The most abundant of these 10 core taxa was an unclassified member of the Alphaproteobacteria class in the order Rhizobiales (OTU2), which accounted for 11.6±6.8% of reads in all phyllosphere samples. At the genus level, the core taxa are affiliated with *Hymenobacter*, *Sphingomonas*, *Terriglobus*, *Spartobacteria genera incertae sedis*, and unclassified genus within the Rhizobiales order. BLASTn analysis of the 10 core OTUs revealed the closest relatives were isolates from diverse environments such as freshwater, soil, and plant material (S10 Fig).

### Habitat specificity of the mānuka phyllosphere microbiome

To ascertain the habitat-specificity of core taxa in the mānuka phyllosphere, the microbial communities in the surface soil surrounding each sample tree were similarly characterised. Across the 29 soil samples, 6,905 OTUs were detected at an average 1820±314 OTUs per sample (S3 Table). In total, 27 bacterial phyla and four archaeal phyla were identified. Taxa belonging to nine bacterial phyla represented the largest proportion of reads: Proteobacteria (29.6%) Acidobacteria (19.7%), Bacteriodetes (13.1%), Verrucomicrobia (9.6%), Actinobacteria (5.5%), Planctomycetes (4.6%), Chloroflexi (2.3%) and Bacteria candidate division WPS2 (1.6%) (S5 Fig).

At the OTU level, phyllosphere and soil community compositions were significantly different, with habitat explaining 41% of the variation in Bray-Curtis dissimilarities (Fig 2A & PERMANOVA on Bray-Curtis dissimilarities, P< 0.001). Notably, a soil core microbiome was not detected. Overall, 599 OTUs were shared by at least one soil and leaf sample, rendering 783 and 6,306 OTUs habitat-specific to the leaf surface and surrounding soil environment, respectively. Of the 599 OTUs shared by soil and phyllosphere communities, 256 exhibited significant differential abundances between habitat types; 152 and 104 were significantly more abundant in soil and phyllosphere samples, respectively (Fig 2B & S8 Table). Nine core phyllosphere taxa were detected in at least one soil sample (S7 Table). However, these nine core taxa were present in soil at very low relative abundances and were significantly enriched in the phyllosphere (Fig 1B and S11 Fig).

### Variation in the mānuka phyllosphere microbiome

Spatial variation in bacterial community structure of the mānuka phyllosphere was explored at the OTU level in a hierarchical manner: at levels of branches, trees, and sites. Phyllosphere communities sampled from the same host tree tended to cluster together and were generally more similar compared to communities sampled from different trees within any given site (S6 and S7 Figs). The effect of host tree explained 25% of variation in the phyllosphere community structure (PERMANOVA on Bray-Curtis dissimilarities, P = 0.001). Phyllosphere communities at different sampling locations formed distinct clusters, and sampling locations explained the largest proportion (50%) of the variation in phyllosphere community structures (Fig 3 & PERMANOVA on Bray-Curtis dissimilarities, P = 0.001). Site-specific community differences were observable even at the phylum level, with increased relative abundances of Firmicutes and Proteobacteria at sites HT and SL respectively (S4 Fig). In comparison, sample site explained only 39% of variation in soil bacterial community structure, and 60% of the variation

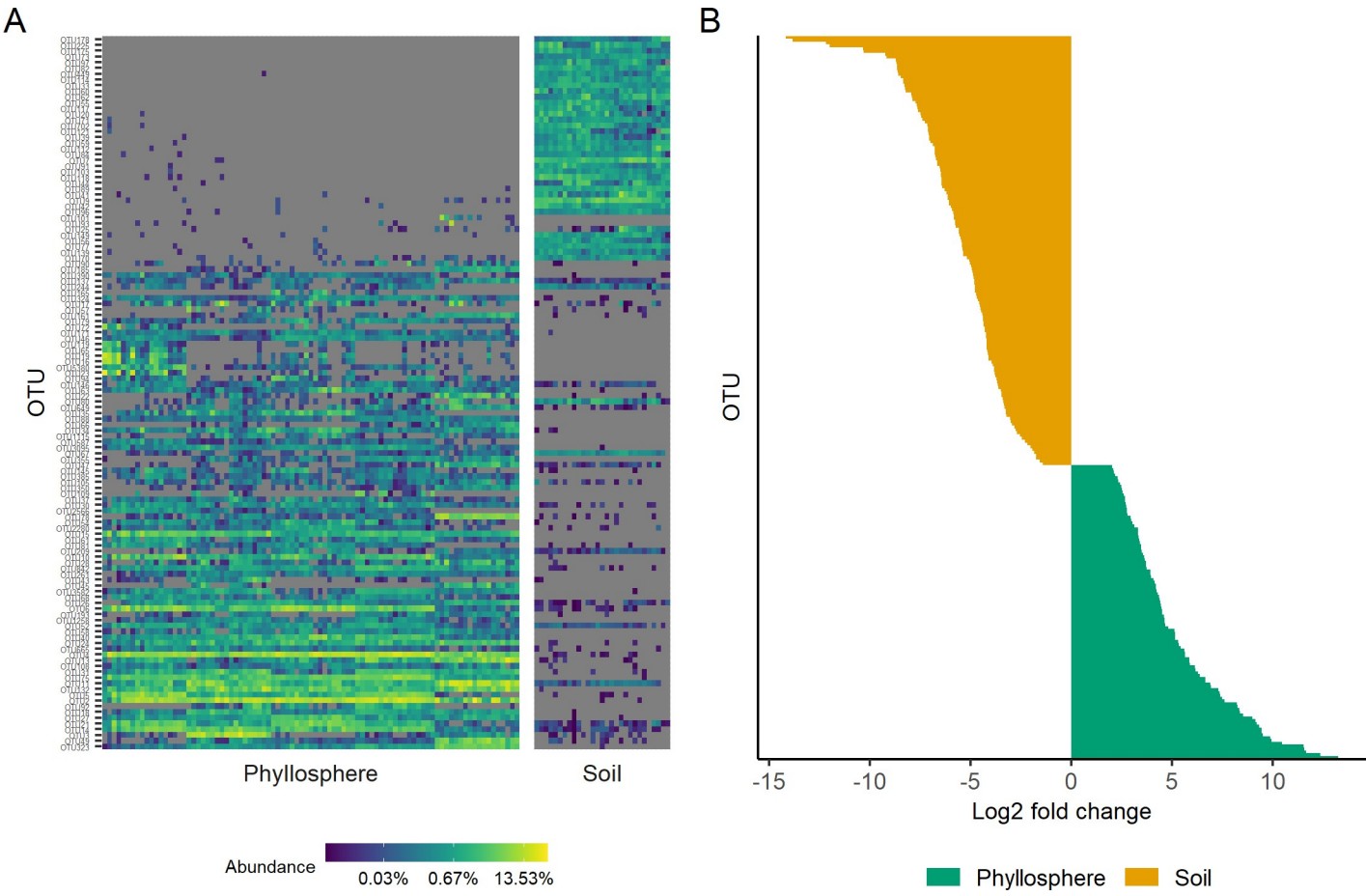

**Fig 2. Differential abundance of taxa in phyllosphere and associated soil communities.** (A) Heatmap depicts the differential relative abundance of OTUs in each environment. Only OTUs with mean relative abundance greater than $1 \times 10^{-3}$ are included. (B) OTUs present in both phyllosphere and soil communities that exhibit significant differential abundances. Positive log2 fold change values represent OTUs with increased abundance in the phyllosphere relative to soil (n = 104) and those with negative log2 fold change values represent OTUs with increased abundance in soil relative to the phyllosphere (n = 152). A full list of OTUs with significant differential abundances, defined by DeSeq2 log-fold difference with an adjusted p value of $\leq 0.01$, is presented in S8 Table.

was attributable to the individual host tree (PERMANOVA on Bray-Curtis dissimilarities, P = 0.001).

### Biogeography

To disentangle the relative effects of sampling location, environmental variables, and host traits on the mānuka phyllosphere microbiome, Mantel and partial Mantel analyses were performed on total, non-core, and core taxa.

Sampling locations were separated by linear distances ranging from 65 to 333 km. On average, samples collected from within the same site were 34% dissimilar, whereas those collected from different sites located 50–150 km and 150–200 km apart were 55% and 72% dissimilar, respectively (Fig 4). Mantel analyses revealed a significant, albeit weak, correlation between total community dissimilarity and geographic distance (rM = 0.15, P = 0.01). This distance-decay relationship was increased in non-core taxa (rM = 0.27, P = 0.002), and diminished in core taxa (P = 0.69) (S9 Table). The mānuka populations selected for sampling spanned a range of elevations from 66 m to 634 m ASL. On average, samples that differed in elevation by less than 100 m exhibited 38% dissimilarity. Samples that differed in elevation by 200–400 m

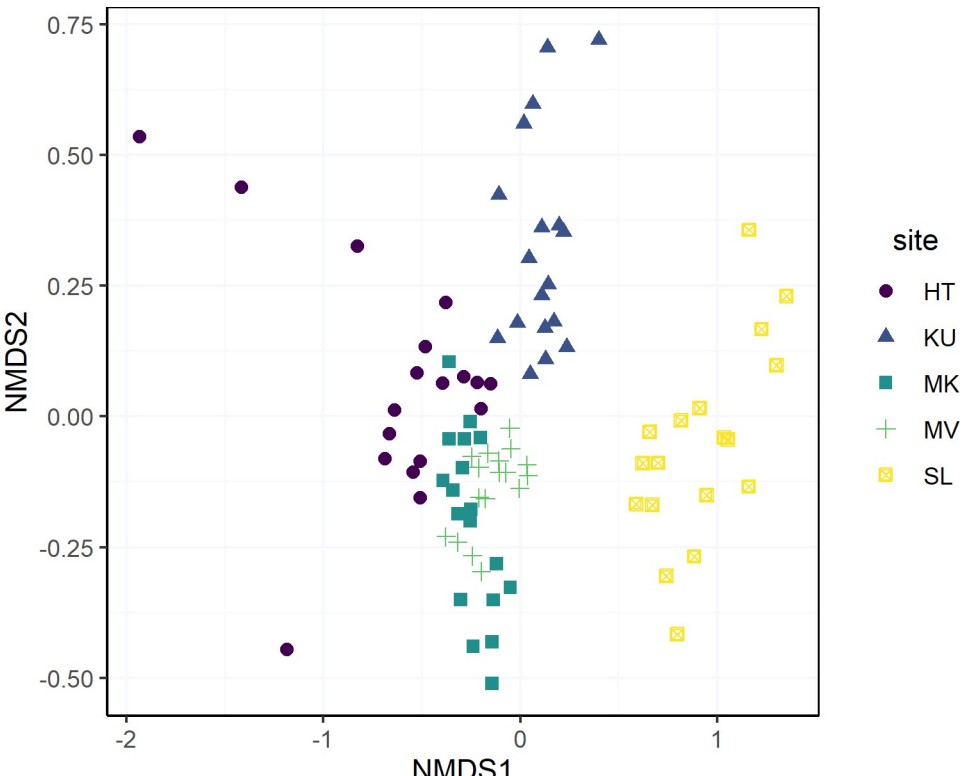

**Fig 3. Nonmetric multidimensional scaling (NMDS) showing the community structure of the mānuka phyllosphere microbiome across sites.** Distance based on Bray-Curtis dissimilarity. The stress value of the NMDS plot is 0.15.

were on average 55% dissimilar. Meanwhile, samples that differed in elevation by more than 500 m were on average 75% dissimilar (Fig 4). Partial Mantel analysis, controlling for spatial autocorrelation, detected a significant relationship between total community dissimilarity and elevation (rM = 0.58, P = 0.001) (S9 Table). Moreover, significant correlations were identified between elevation and both non-core (rM = 0.60, P = 0.001) and core taxa (rM = 0.43, P = 0.001) (S9 Table). Community dissimilarity of total, non-core, and core taxa also exhibited significant correlations with latitudinal and longitudinal distances (S9 Table).

Of the 14 environmental variables explored, partial Mantel analyses found that average night temperature (rM = 0.78, P = 0.001), day-night temperature differential (rM = 0.65, P = 0.001), monthly precipitation (rM = 0.58, P = 0.001), monthly cloud cover (rM = 0.60, P = 0.001), and monthly sun hours (rM = 0.61, P = 0.001 were most strongly correlated with community dissimilarity (S8 Fig and S10 Table). Average day temperature, relative humidity, monthly pressure, and monthly humidity also demonstrated a significant, albeit weaker, correlation with community dissimilarity (S10 Table). Community dissimilarity of non-core taxa showed strong correlations (rM > 0.5) with average night temperature (rM = 0.80, P = 0.001), day-night temperature differential (rM = 0.62, P = 0.001), monthly precipitation (rM = 0.62, P = 0.001), monthly cloud cover (rM = 0.60, P = 0.001), and monthly sun hours (rM = 0.61, P = 0.001) (S9 Fig). In contrast, community dissimilarities of core taxa were strongly correlated with average night temperature (rM = 0.63, P = 0.001) and exhibited weaker associations with other environmental parameters (S9 Fig and S10 Table).

A significant correlation was found between total community dissimilarity and tree height, whereby trees of different heights tended to host different communities compared to trees of

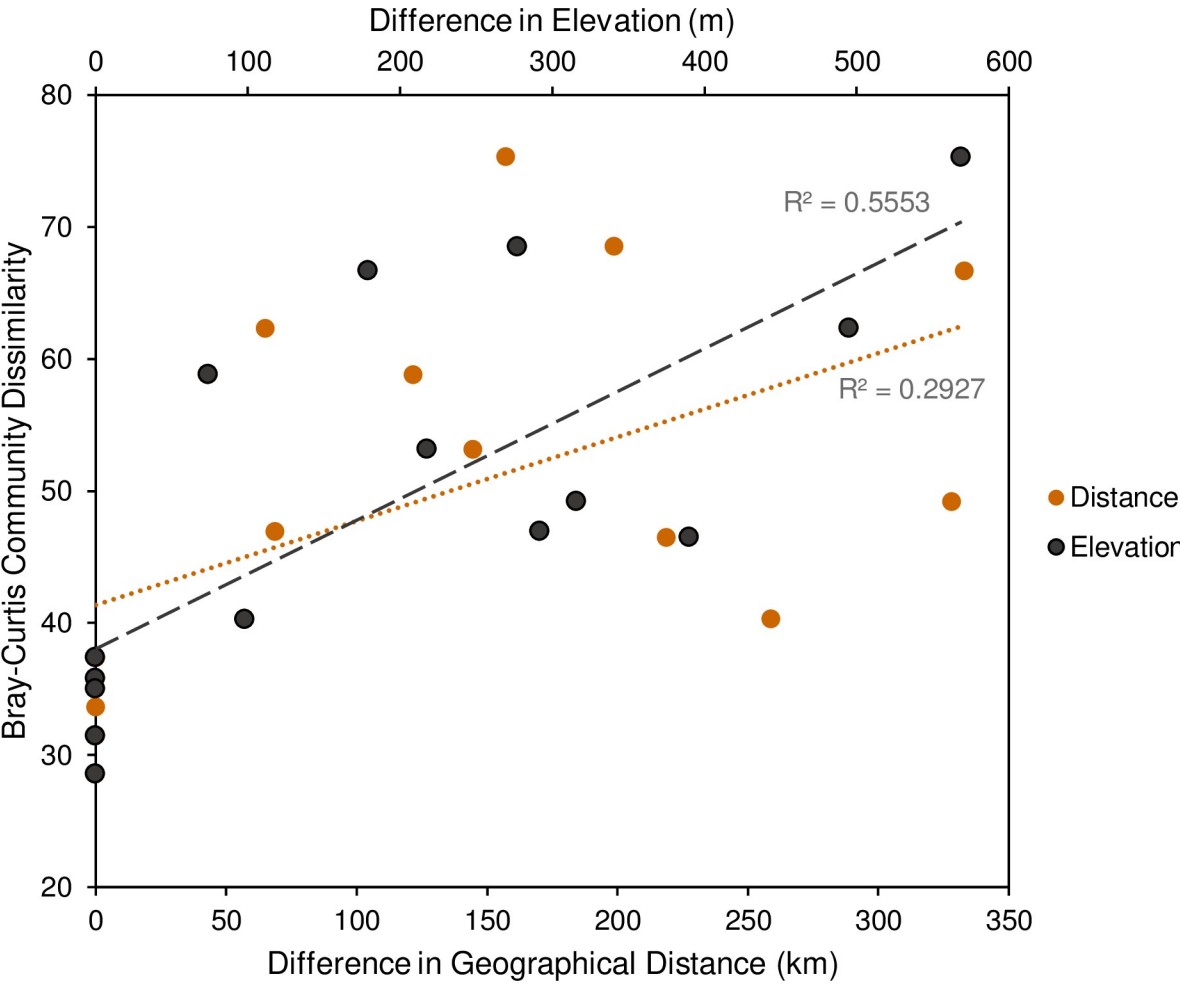

**Fig 4. Mānuka phyllosphere community dissimilarity increases with increasing geographic distance and elevation between sample regions.** Community dissimilarity is based on the Bray-Curtis dissimilarity index averaged between sample regions.

similar heights (rM = 0.19, P = 0.022). No significant correlation was detected between total community dissimilarity and tree diameter, branch aspect, or branch height (S11 Table). Non-core taxa also exhibited a significant correlation with tree height (rM = 0.23, P = 0.004), which was not observed in core taxa (S11 Table). No correlation was found between the community dissimilarity of either non-core or core taxa and tree diameter, branch aspect, and branch height (S11 Table).

### Relative importance of spatial, climatic and host tree factors

Variation partitioning was used to further resolve the relative contributions of environmental variables, spatial factors, and host traits on the total phyllosphere community structure. In total, 42% of the phyllosphere community variation was explained by these three groups of explanatory variables (Fig 5). Environmental variables had the greatest impact on the observed heterogeneity in phyllosphere community structure, independently explaining 9% of the variability. Geographical space also had a significant and independent impact on the variability of community structure, corroborating correlational analyses described above. Together, the combined effect of both environmental and spatial explanatory variables accounted for 21% of

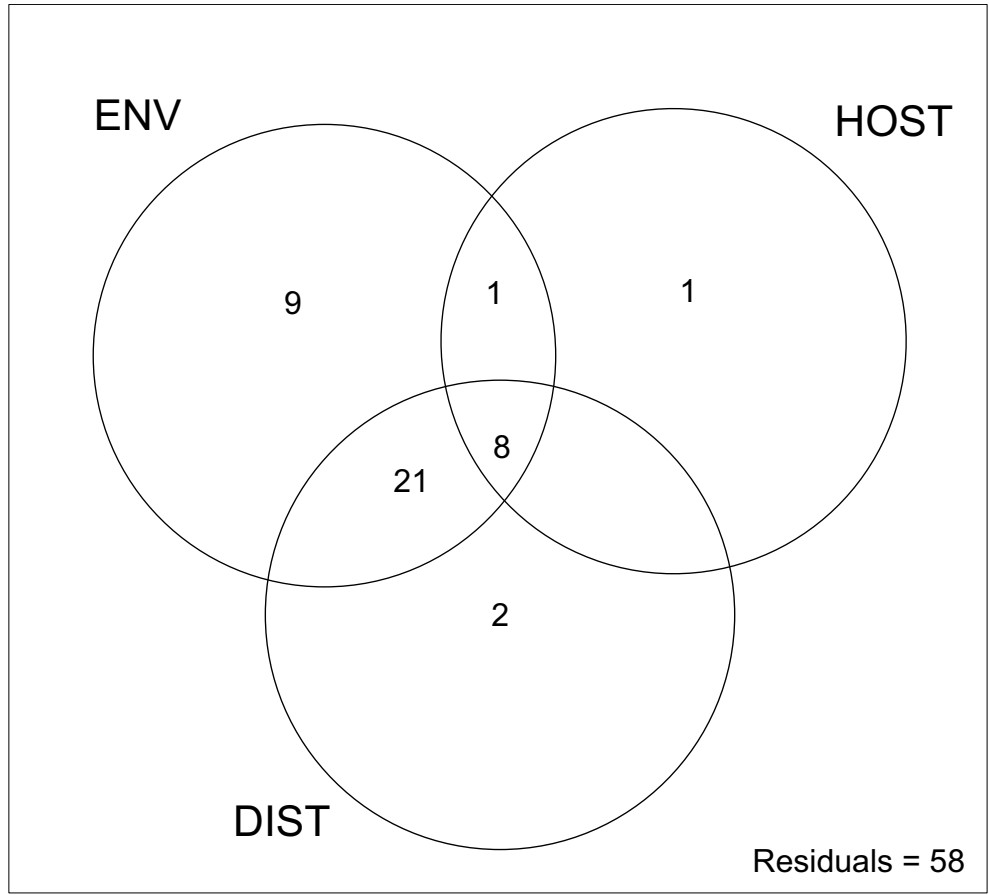

**Fig 5. The proportion (%) of mānuka phyllosphere microbiome variation explained by environmental, spatial, and host tree variables.** Results of partial regression analysis show the proportion of variation attributable to each individual and combination of explanatory variables. Environmental variables (ENV) represent the night-day temperature differential, night temperature, monthly precipitation, and monthly cloud cover. Spatial variable (DIST) represents the straight-line distance between sample coordinates. Host tree variables (HOST) represent tree height and tree diameter.

the total variation in phyllosphere community structure. However, the independent effect of geographical space was relatively small, explaining only 3% of the variability in bacterial community structure independent of other explanatory variables. The relative effects of measured host traits, both independently and in combination with climatic variables, were very small, explaining only a further 1% of observed variation. Collectively, host tree-related traits, environment, and spatial explanatory factors accounted for 8% of the total variation.

## Discussion

The phyllosphere microbiome is increasingly recognised as an influential component of host plant biology [5–7]. However, the strength and stability of specific bacterial associations in the phyllosphere remains unclear. Here, we provide the first characterisation of the bacterial communities in the phyllosphere microbiome of *Leptospermum scoparium* (mānuka) across five distinct and distant populations in the North Island of New Zealand. Across replicate samples and with the most stringent criterion (presence in all samples), we identified a core microbiome in the collective mānuka phyllosphere. This finding is significant, given the geographical separation and environmental heterogeneity of the sampled populations, and provides

evidence in support of our hypothesis that the mānuka phyllosphere microbiome exhibits specific host association.

## The mānuka phyllosphere core microbiome

The core phyllosphere microbiome (10 OTUs) represented a large proportion of the total phyllosphere community yet was either rare or entirely absent in surrounding surface soil, suggesting specialisation of these microbial taxa to the habitat or strong selection by the host plant. At the phylum level, the core microbiome comprised major lineages of bacteria also identified in the phyllosphere of other plant species [10, 11, 13]. Four of the most abundant core OTUs were affiliated with the Rhizobiales order (Alphaproteobacteria). OTUs related to order Rhizobiales have frequently been identified in the phyllosphere of temperate forest tree species [70, 71]. Other core OTUs were assigned to genera not commonly reported in the phyllosphere, such as *Terriglobus*, which are generally considered well adapted to survive oligotrophic environments [72]. Many core OTUs also represented previously undescribed microorganisms, indicating a diversity within these phyllosphere bacterial lineages that may be unique to mānuka. The apparent host-specific association in the mānuka phyllosphere microbiome supports previous studies that identified species-specific communities in the phyllosphere of co-occurring plant species [13]. Genotype-dependent characteristics of leaf chemistry and morphology have been suggested to drive such host-specific patterns [17]. Notably, medicinal plants have previously been reported to harbour highly specialised microbial communities due to their unique and structurally divergent secondary metabolites [73]. The unique components of mānuka's antimicrobial essential oils may therefore play a role in the selection and maintenance of specific bacterial lineages in its phyllosphere across large geographical ranges and variable environments [74].

In accordance with the prevailing core microbiome concept, a stable association between the mānuka host and a habitat-specific group of microorganisms suggests these taxa may be functionally important for the integrity of the phyllosphere microbiome and host plant [20, 22]. Consistent with this notion, multiple members affiliated with the core bacterial genera identified here are known to associate closely with plants. For example, *Sphingomonas* has been shown to enhance the host plant immune response towards pathogenic strains of *Pseudomonas* [18]. Rhizobiales also contains many genera of methylotrophic bacteria that metabolise methanol and enhance plant growth by producing phytohormones, such as indole-3-acetic acid [75, 76]. However, given the large strain-level functional diversity of microorganisms and that 16S rRNA gene sequence-based inference of potential functions are highly problematic, the functional roles of these core taxa remain an open question [77].

## Biogeographic patterns differ between core and non-core members of the mānuka phyllosphere

Besides harbouring a core microbiome, the mānuka phyllosphere exhibited biogeographical patterns that appeared to be shaped by both deterministic and neutral processes [78, 79]. Biogeographical patterns have been identified in the phyllosphere of other tree species [15, 80]. However, few studies have examined the biogeographical patterns of different microbial populations in the phyllosphere (e.g., rare versus abundant and core versus non-core taxa) as in other environments [16, 45, 81]. Our results revealed that biogeographical patterns were strongest in non-core taxa, exhibiting a non-random distribution across space and environmental gradients that contrasted with the cosmopolitan distribution of the relatively dominant core microbiome. By virtue of their definition, non-core taxa do not consistently associate with a host species, and their presence and abundance are therefore hypothesised to be

predominantly structured by environmental selection and/or stochastic processes [20, 82]. The partial Mantel test revealed that environmental conditions, such as night temperature and precipitation, and spatial factors, such as elevation, played significant roles in shaping the composition of non-core taxa in the mānuka phyllosphere. Interestingly, a distance-decay relationship was also identified exclusively in non-core taxa, illustrating a potential role of dispersal in the assembly of the non-core community [83, 84]. Consistent with this finding, a distance-decay relationship has been previously identified in rare taxa of the *Tamarix* phyllosphere [16]. Together, these results illustrate the potential of the mānuka phyllosphere microbiome to exhibit both environmental segregation and biogeographic provincialism in the non-core community.

### The mānuka phyllosphere microbiome is not stochastically recruited from soil

Significant differences in community composition were observed between the host microbiome in the phyllosphere and free-living communities in the surrounding soil. As expected, soil bacterial communities were more diverse than those in the phyllosphere. Given the environmental heterogeneity of the surrounding environments in which we sampled, the collection of neutral and deterministic forces shaping the soil communities were likely very diverse. Notably, taxa that were abundant in the phyllosphere, including the core microbiome, were essentially absent in surrounding soil. This finding contrasts with previous studies that report a large proportion of abundant phyllosphere taxa are also present in soil [14, 85]. Although the major reservoir of the mānuka phyllosphere microbiome cannot be directly inferred from our analysis, if phyllosphere bacterial communities were assembled stochastically from the surrounding environment, the diversity of the surrounding soils would have been expected to drive large regional differences as observed in other plant species, such as the grape vine [14]. Instead, the convergence consistently observed in the phyllosphere microbiome independent of the surrounding environment suggests that soil has limited influence on the community structure of the mānuka phyllosphere microbiome.

### Future directions

Our study was strengthened by aspects of our methodological design that allowed us to essentially exclude cross-contamination as an alternative explanation for the presence of a ubiquitous core microbiome. Specifically, soil samples were processed simultaneously with leaf samples to mitigate the risk of cross-contamination during sample collection or processing, and fusion primers were used to prepare 16S rRNA gene PCR amplicons in a one-step protocol to eliminate the risk of cross-contamination by aerosolised PCR products [47–51]. The immediate next step should be to investigate the functional potential and role of the core phyllosphere microbiome in potentially mediating the physiological properties of mānuka using a combination of shotgun metagenomic sequencing and cultivation approaches. Metagenomics will also reveal the diversity and abundance of micro-eukaryotes such as fungi and protists, which are increasingly recognised as potential determinants of plant-microbe interactions and host fitness [8]. Incorporating host genetic analyses and co-occurring plant species into future analyses will also shed light on the relationship between the mānuka phyllosphere microbiome and host genetics as well as the role of recruitment in community assembly.

### Conclusion

Our findings demonstrate that *Leptospermum scoparium* (mānuka) possesses a dominant core phyllosphere microbiome that is persistent across geographically distant populations. Core

phyllosphere taxa represented a large proportion of the total phyllosphere community yet were rare or entirely absent in surrounding soils, providing support for hypothesised specific host association and potentiating the role of host selection of functionally important microorganisms. Strong biogeographical patterns were also identified in the mānuka phyllosphere, indicating that spatially structured environmental gradients may play a role in shaping the composition of non-core microbiota. Together, these findings illustrate that the mānuka phyllosphere microbiome exhibits complex biogeography that likely resulted from divergent processes driving the assembly of core and non-core microorganisms. Understanding potential roles of the mānuka phyllosphere microbiome in the physiology of the nationally and culturally treasured indigenous plant will be of wide scientific and general interest. Our findings also illustrate the usefulness of the mānuka phyllosphere microbiome for expanding our knowledge of specific host-microbe associations in the phyllosphere and support the notion that investigation of phyllosphere microbiomes should be conducted as per our *senso stricto* definitions of the phyllosphere and core microbiome to ensure relevance and generalisability of findings.

## Supporting information

**S1 Fig. Location of *Leptospermum scoparium* (mānuka) populations in the North Island of New Zealand.** Borders represent regional boundaries. Total rain (mm) and total sun (mm) data retrieved from the National Climate Database (NIWA) for the month prior to sampling. Map made with Natural Earth.
(PDF)

**S2 Fig. Environmental conditions prior to sampling.** Air temperature, relative humidity, and photosynthetically active radiation was measured at 15-minute intervals with a datalogger across the 24 h period prior to sampling. Each site is represented by colour.
(PDF)

**S3 Fig. Alpha diversity of phyllosphere communities from different sample sites.** (a) Observed richness (ANOVA, $p = 2.99 \times 10^{-6}$, F = 9.26) was significantly higher at KU and MV compared to other sites. (b) Chao1 (ANOVA, 0.0023, F = 4.53) was significantly higher at KU compared to sites HT, MK, and MV.
(PDF)

**S4 Fig. Average relative abundance of phyla in the phyllosphere microbiome.** Bars represent the average relative abundance per sample tree (n = 3) and are grouped by site (HT, KU, MK, MV, SL). Phyla are represented by colour.
(PDF)

**S5 Fig. Relative abundance of phyla in the soil microbiome surrounding mānuka trees.** Bars represent one sample. Samples are grouped by site (HT, KU, MK, MV, SL). Phyla are represented by colour.
(PDF)

**S6 Fig.** Mānuka phyllosphere community dissimilarity within (A) and between (B) trees at each sample site. Community dissimilarity is based on Bray Curtis dissimilarity values. Boxes represent the interquartile range dissimilarity values. The thick bars represent the median of dissimilarity values, and the vertical segments extend to the fifth and the 95th percentiles of the distribution of values. Wilcoxon test statistically significant differences are shown with asterisks ($^{*}p<0.05$, $^{**}p<0.01$, $^{***}p<0.001$, $^{****}p<0.0001$).
(PDF)

**S7 Fig. Hierarchical clustering analysis of the mānuka phyllosphere microbiome.** Branches are coloured according to sample region and sample date. Boxes indicate five cluster groups (k). Branch labels correspond to individual treeID. The length of the branches corresponds with Bray-Curtis dissimilarity between samples.
(PDF)

**S8 Fig. Correlation between mānuka phyllosphere total community dissimilarity and environmental dissimilarity.** Community dissimilarity (y axis) is based on Bray Curtis. Environmental dissimilarity (x axis) is based on Euclidean distances of average night temperature (A), day-night temperature differential (B), monthly rain (C), monthly cloud cover (D), and monthly sun hours (E). Line represents Pearson product moment correlation coefficient (R).
(PDF)

**S9 Fig. Correlation between mānuka phyllosphere community dissimilarity and environmental dissimilarity for core and non-core taxa.** Community dissimilarity (y axis) of non-core (left panels) and core (right panels) taxa is based on Bray Curtis. Environmental dissimilarity (x axis) is based on Euclidean distances of average night temperature (AB), day-night temperature differential (CD), monthly rain (EF), monthly cloud cover (GH), and monthly sun hours (IJ). Line represents Pearson product moment correlation coefficient (R).
(PDF)

**S10 Fig. Phylogenetic maximum-likelihood analysis based on 16S rRNA gene PCR amplicon sequences of core phyllosphere OTUs and bacterial isolates with the closest sequence identity.** Isolates were identified using BLASTn; the accession number and isolation source of each isolate is provided. Sequences were aligned with ClustalW. Bootstrap probabilities based on 1000 replications are shown. Bar: 0.05 nucleotide substitutions rate (Knuc) units.
(PDF)

**S11 Fig. OTUs with significantly increased abundance in the phyllosphere relative to soil (n = 104).** Estimates of log2 fold changes were obtained using DESeq2. Colour depicts members of the core phyllosphere microbiome. A full list of OTUs with significant differential abundances, defined by a DeSeq2 log-fold difference with an adjusted p value of $\leq 0.01$, is presented in S8 Table.
(PDF)

**S1 Table. Spatial and environmental metadata for each mānuka sample site.** Monthly averages were obtained from the National Climate Database (NIWA). Other environmental conditions were measured at 15-minute intervals with a datalogger across the 24 h period prior to sampling.
(PDF)

**S2 Table. Spatial metadata for each mānuka phyllosphere sample.**
(PDF)

**S3 Table. Number of samples, reads, and OTUs for total community, phyllosphere samples, and soil samples.**
(PDF)

**S4 Table. Mānuka phyllosphere microbiome raw alpha diversity values for observed richness, Shannon, and Chao1 indices.**
(PDF)

**S5 Table. Site-wise Tukey test for mānuka phyllosphere microbiome richness and Chao1.** Statistically significant differences are shown with asterisks (*p<0.05, **p<0.01, ***p<0.001, ****p<0.0001).
(PDF)

**S6 Table. Correlation with mānuka phyllosphere microbiome alpha diversity and environmental parameters.** Correlations calculated using Pearson's product-moment correlation coefficient test (Corr). p > 0.5 correlation values are not shown.
(PDF)

**S7 Table. Relative abundance of core taxa across all mānuka phyllosphere (n = 89) and soil (n = 29) samples.** Relative abundance of core taxa in phyllosphere samples is averaged per tree and site (mean ± SD).
(PDF)

**S8 Table. Full list of OTUs with significant differential abundances, defined by DeSeq2 log-fold difference with an adjusted p value of ≤ 0.01.**
(XLSX)

**S9 Table. Mantel and Partial Mantel test correlations between mānuka phyllosphere community dissimilarity and spatial variables.** Correlations are performed on Bray Curtis dissimilarity of total, non-core, and core taxa and Euclidean distances of spatial parameters using Pearson's product-moment correlation coefficient test (Corr). Positive correlations greater than 0.5 are underlined.
(PDF)

**S10 Table. Partial Mantel test correlations between mānuka phyllosphere community dissimilarity and Euclidean distances of environmental variables.** Correlations are performed on Bray Curtis dissimilarity of total, non-core, and core taxa and Euclidean distances of environmental parameters using Pearson's product-moment correlation coefficient test (Corr). Positive correlations greater than 0.5 are underlined. The symbol (m) represents monthly averages obtained from the NIWA database. Other environmental conditions were measured with a datalogger at each site every 15-minute intervals across the 24 h period prior to sampling.
(PDF)

**S11 Table. Partial Mantel test correlations between mānuka phyllosphere community dissimilarity and Euclidean distances of host variables.** Correlations are performed on Bray Curtis dissimilarity of total, non-core, and core taxa and Euclidean distances of host tree parameters using Pearson's product-moment correlation coefficient test (Corr).
(PDF)

## Acknowledgments

We thank Steens Honey, East Taupō Lands Trust, Timberlands Ltd, and Waipa District Council for providing and assisting with access to sites. Roanna Richards, John Longmore, and Shaun Sanders provided assistance in the laboratory.

## Author Contributions

**Conceptualization:** Charles K. Lee.

**Data curation:** Anya S. Noble.

**Formal analysis:** Anya S. Noble, Charles K. Lee.

**Investigation:** Anya S. Noble, Stevie Noe.

**Methodology:** Anya S. Noble, Stevie Noe, Michael J. Clearwater, Charles K. Lee.

**Project administration:** Anya S. Noble, Charles K. Lee.

**Resources:** Stevie Noe, Michael J. Clearwater, Charles K. Lee.

**Supervision:** Charles K. Lee.

**Validation:** Anya S. Noble.

**Visualization:** Anya S. Noble.

**Writing – original draft:** Anya S. Noble.

**Writing – review & editing:** Charles K. Lee.

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
