## [Decision Letter · Decision Letter 0]

2 Mar 2020

PONE-D-20-02828

A core phyllosphere microbiome exists across distant populations of a tree species indigenous to New Zealand

PLOS ONE

Dear Dr. Lee,

Thank you for submitting your manuscript to PLOS ONE. After careful consideration, we feel that it has merit but does not fully meet PLOS ONE’s publication criteria as it currently stands. Therefore, we invite you to submit a revised version of the manuscript that addresses the points raised during the review process.

The manuscript was reviewed by two experts in the field . Both raised some important points which I believe can be addressed in a revised version.  

We would appreciate receiving your revised manuscript by Apr 16 2020 11:59PM. To enhance the reproducibility of your results, we recommend that if applicable you deposit your laboratory protocols in protocols.io, where a protocol can be assigned its own identifier (DOI) such that it can be cited independently in the future. For instructions see: http://journals.plos.org/plosone/s/submission-guidelines#loc-laboratory-protocols

We look forward to receiving your revised manuscript.

Kind regards,

Omri Finkel, PhD

Academic Editor

PLOS ONE

Journal Requirements:

2. In your Methods section, please provide additional information regarding the permits you obtained for the work. Please ensure you have included the full name of the authority that approved the collections sites access and, if no permits were required, a brief statement explaining why.

3. We note that you are reporting an analysis of a microarray, next-generation sequencing, or deep sequencing data set. PLOS requires that authors comply with field-specific standards for preparation, recording, and deposition of data in repositories appropriate to their field. Please upload these data to a stable, public repository (such as ArrayExpress, Gene Expression Omnibus (GEO), DNA Data Bank of Japan (DDBJ), NCBI GenBank, NCBI Sequence Read Archive, or EMBL Nucleotide Sequence Database (ENA)). In your revised cover letter, please provide the relevant accession numbers that may be used to access these data. For a full list of recommended repositories, see http://journals.plos.org/plosone/s/data-availability#loc-omics or http://journals.plos.org/plosone/s/data-availability#loc-sequencing.

4. We note that Figure S1 in your submission contain map images which may be copyrighted. All PLOS content is published under the Creative Commons Attribution License (CC BY 4.0), which means that the manuscript, images, and Supporting Information files will be freely available online, and any third party is permitted to access, download, copy, distribute, and use these materials in any way, even commercially, with proper attribution. For these reasons, we cannot publish previously copyrighted maps or satellite images created using proprietary data, such as Google software (Google Maps, Street View, and Earth). For more information, see our copyright guidelines: http://journals.plos.org/plosone/s/licenses-and-copyright.

1.    You may seek permission from the original copyright holder of Figure S1 to publish the content specifically under the CC BY 4.0 license. 

Reviewers' comments:

Reviewer's Responses to Questions

**Comments to the Author**

1. Is the manuscript technically sound, and do the data support the conclusions?

Reviewer #1: No

Reviewer #2: No

2. Has the statistical analysis been performed appropriately and rigorously? 

Reviewer #1: Yes

Reviewer #2: No

3. Have the authors made all data underlying the findings in their manuscript fully available?

Reviewer #1: Yes

Reviewer #2: Yes

4. Is the manuscript presented in an intelligible fashion and written in standard English?

Reviewer #1: Yes

Reviewer #2: Yes

5. Review Comments to the Author

Reviewer #1: Anya Noble, Charles Lee

PlosOne

The authors investigated microbial communities on five Manuka populations in New Zealand using 16S amplicon sequencing. They contrasted the phyllosphere community composition from the composition of surface soil underneath the trees.

The english used in the publication is good and the sequencing and formal analysis is of high standard.

The claimed hypothesis that antimicrobial activity of honey is related to the phyllosphere microbiome cannot be answered by the analysis and experiments performed. If the authors were to address this question, they would have to measure the activity of honey associated with individual plants and determine the phyllosphere microbiome of those plants too.

For arguments sake, why would antimicrobial activity be related to phyllosphere microbes compared to endophytes or the rhizosphere microbiome? This is not addressed at all.

In a nutshell, I believe that the authors should reformulate their hypothesis and question so that they can actually test them with their dataset. I suggest to focus on geographical location and latitude instead.

I find that the argument that a core microbiome would be responsible for the medicinal/ antimicrobial activity flawed. Wouldn’t fluctuations of this activity better be explained by the changes in accessory microbiota?

As a general note, the use of references is excessive - not every fact needs three references in the introduction. Also in the introduction, the authors go back and forth between mammalian and plant references a lot - since this is a study relating to plants and all the concepts they are referring too are well established, they should focus on those rather than mammalian studies. E.g. the concept of core microbiome is well studied in the phyllosphere. This will streamline and focus the study while avoiding overstatements.

As another general note - many references are not appropriately chosen and should be double checked - by just looking into a few I found many that were not cited in a correct context. Going through all is too big of a job for this already extensive review.

The introduction could be streamlined by reorganising the lines of thoughts. Currently, it goes back and forth between different lines of thoughts. The authors will have to do their due diligence and carefully check everything themselves.

Lines 51-55 I think these are big words - they need more explanation and references. “Must” is imperative and it is unclear to me what the authors are referring too when they assert that this “must” be done. Plant leaves and other plant organs have previously been shown to exude nutrients. Plant leaves have species specific physicochemistries and morphologies etc. The existents of a core microbiome does not necessarily relate to host-microbe interactions, even though it is tempting to speculate the latter.

Drivers of plant microbiotas have been reviewed elsewhere - I suggest the authors use these reviews to support their claims.

Line 56 The phyllosphere is often troublesome to define and different definitions can be found - refer to the 2019 review of Johan Leveau. Most often it has been defined as the microbial habitat that encompasses the above-ground surface of plants and is contrasted by the rhizosphere.

The authors spend many words on core-microbiome while also stating that conflicting information exists for some plant hosts. Doesn’t that already suggest that the concept as the authors want to use it does not hold? This seems like a circular argument. It has been claimed before that the phyllosphere microbiome of plants is compositionally consistent at a low phylogenetic level - i.e. different species of the same genus can be found in different environments and in different years. Will this change you assessment in lines 61-66?

The authors missed to cite the study of Wicaksono et al. (10.1371/journal.pone.0163717) who were in my knowledge the first who investigate the Manuka endosphere microbiome. The here presented study should be compared and contrasted with this previous study.

Next to the study above - it is not discussed how unique the manuka microbiota is - please compare and contrast from other phyllosphere studies.

I am surprised that top soil microbiome was used to contrast the phyllosphere samples - this is well known to be distinct from leaf microbiomes and rhizosphere microbiomes. This does not seem a suitable comparison. Instead, other plants in the surrounding would have been useful.

The authors investigated the differences between geographical as well as elevations, it is unclear to me though if both factors can be separated as the different manuka host populations were found at different elevations and thus distance and elevation are connected.

The authors sampled at different times - connected to flowering of the Manuka, did they analyse the effect of sampling date on communities?

I do not understand the paragraph connected to figure 5 or figure 5 itself - this seems overanalysed to me. Phyllosphere communities can be diverse, recent studies indicate that priority effect can have strong influence on the communities - the authors seem to infer that every plant had the same founder population of bacteria surrounding them. The authors have not included host genetic in this analysis, why? Many others have found that this can impact on the phyllosphere microbiome (e.g. Bodenhausen et al. 2014)

The PICRUSt analysis seems another step of over analysing the data - what is the purpose of this within the study as a whole? If anything, it would have been useful to try and identify genes that may be responsible for changes in the amount of active compounds in the Manuka honey to attempt the provide evidence for the authors initial hypothesis. Here it is merely used to contrast leaf from soil - which has been achieved by the 16S diversity already and does not add new information.

Discussion line 410: I agree with the claim made, but the references used seem off - there is studies that show that phyllosphere bacteria impact on the host directly e.g. Vogel et al. 2016 (DOI: 10.1111/nph.14036).

The authors claim that they are the first to identify a core microbiome despite that others have performed similar studies with similar tools. Does that mean that Manuka is different compared to many other plants and that the results here cannot be generalised over many other plant species? I would think so - but I do not read in the provided manuscript.

Line 429 references 80 and 81 do not fit here.

Line 432 since the sentence should relate to the phyllosphere the reference 83 cannot be used as it studies endophytic microbes

Line 442-444 Sphingomonads in the study did not actively antagonise Pseudomonas - in fact refer to the Vogel paper further up - apparently Sphingomonas Fr1 activate plant immune responses which allowed for a faster response of the plant towards the pathogen, also Sphingomonads can also be plant pathogens Buonaurio et al. 2002 (DOI: 10.1099/ijs.0.02063-0), so this claim might not hold.

The cross-contamination paragraph is odd and uncommon to see in such a manuscript - it is better suited in a supplement or in a protocol paper. I suggest to delete it from the main manuscript.

After reading the article, I want to learn more about the core-microbiome of manuka - but the authors are not addressing this question - what organisms can we find across Manuka, are they common on other plants, what are their likely properties, do they have plant growth promoting properties, are they plant pathogens in other systems, or antagonists etc.?

Reviewer #2: In this manuscript, the authors measured the leaf and soil microbial communities across five distinct populations of Leptospermum scoparium (total of 29 individual tree sampled, ~ 6 trees per site, three branches per tree), an indigenous tree species to NZ. The authors hypothesize that the phyllosphere microbiota plays a role in mediating host physiology and thus exhibits specific host association. They identify a habitat specific and abundant leaf core microbiome present across all samples and claim that this result supports their hypothesis that phyllosphere microorganisms exhibit host association and potentially mediate physiological traits. This manuscript discusses the relevance and the definition of a core microbiome, which provides pertinent information to the field of host-microorganism interactions. However, the fact that the authors excluded voluntarily the endophytic communities by modifying their protocol make me doubt the accuracy of their claim to have identified microorganisms that mediate plant physiological traits as endophytes are recognized to be key organisms interacting with plant local and systemic immune system. The exploration of biogeography patterns in leaf bacterial communities for a single tree species contributes relevant information to the field.

Introduction: The authors should discuss or at least acknowledge the limits of describing only leaf bacterial communities when micro-eukaryotes such as fungi and protists but also phages are increasingly recognized to play a key role as determinants of plant-microbe interactions and host fitness.

Methods: Why use surface soil and not root associated microorganisms?

Figure S1. Would it be possible to add more information to this figure by overlaying changes in topography, or forest/land type, mean temperature, etc.

Figure 1. Panel A. It would be relevant to have a third panel equivalent to panel A where relative abundance is collapsed per site to present the mean relative abundance of core OTUs because there seems to be site specific trends. Panel B can be rescaled to provide better resolution by having 0.5% as the maximum on the y axis. As of now we barely see anything.

Lines 279-280: How much variation in BC dissimilarity did habitat (leaf vs. soil) explain?

Lines 284-285: It is intriguing that no soil core microbiome was detected, what could explain this?

Lines 368-370: I don’t understand what it means to “represent independently and collectively 42% of the variation in community structure”. Also, are the numbers in Fig. 5 representing %? Is so this should be indicated in the figure or stated in the legend.

Line 448 (and other places): I would refrain from using the word “patterning” as this term is rarely used (to not say never). Please use “patterns” or “dynamics” to use literature-relevant terminology.

Lines 480-483: This is shocking to me. Excluding the leaf endophytes might have dramatic consequences as they are expected to be agents interacting directly with the plant host. Please discuss the relevance of excluding endophytes and the limits this imposes on your study.

Corss-contamination: It seems to me that the section on cross-contamination rather belong in the methods.

Lines 492-495: This also has consequences for the literature where most papers have suggested that soil is a reservoir for leaf microorganisms. Please compare your results with corresponding references that have found opposite results to yours and cite these references.

Lines 500-503: It seems to me that the immediate next step is rather to confirm if this leaf core microbiome is truly playing a role for its host fitness and phenotype.

Figure 3. I find it strange to see that hierarchical cluster was able to perfectly separate samples per site provenance based on bacterial community structure. I would expect that stochasticity would at least have mixed some samples into different site provenance. Could you provide an NMDS/PCoA showing how clearly these five clusters separate based on community structure?

6. PLOS authors have the option to publish the peer review history of their article (what does this mean?). If published, this will include your full peer review and any attached files.

Reviewer #1: Yes: Mitja Remus-Emsermann

Reviewer #2: Yes: Isabelle Laforest-Lapointe

---

## [Author Response · Author response to Decision Letter 0]

14 May 2020

Please see the Cover Letter for Resubmission

---

## [Decision Letter · Decision Letter 1]

27 May 2020

PONE-D-20-02828R1

A core phyllosphere microbiome exists across distant populations of a tree species indigenous to New Zealand

PLOS ONE

Dear Dr. Lee,

Thank you for submitting your manuscript to PLOS ONE. After careful consideration, we feel that it has merit but does not fully meet PLOS ONE’s publication criteria as it currently stands. Therefore, we invite you to submit a revised version of the manuscript that addresses the points raised during the review process.

We look forward to receiving your revised manuscript.

Kind regards,

Omri Finkel, PhD

Academic Editor

PLOS ONE

Additional Editor Comments (if provided):

For this second round of reviews, only one of the original reviewers has agreed to proceed with the revision. As you can see, this reviewer still has some important concerns. In the absence of another reviewer I have stepped in and I have some additional comments:

- I think for the core taxa analysis a slightly more sophisticated approach could be taken. The different samples are not rarefied and you completely discount relative abuncance. The combination of both of these factors could distort your selection criteria. I would suggest to (a) rarefy your samples to a uniform size for this analysis and (b) plot relative abundance against site prevalence (in how many sites is this OTU detected). you can draw lines to demarkate your selection criteria, but it will give the reader an idea of how many OTUs were very close but out of the bounds of your criteria.

- in line 303 you mention you used blastn to identify the taxonomy of your strains. It is not clear to me why you needed to do this considering the taxonomy was already assigned using RDP, which is a superior way to assign taxonomy to 16S reads. If you would like to explore the taxonomic neighborhood of these taxa, I would build a phylogenetic tree out of close RDP or blast hits and place your sequenceds within it.

- since you've sequenced all of these soil samples, I think it would be more interesting if you ran a model to compare soil vs phyllosephre for each OTU rather than simply reporting on the overlap. That way you can also find our whether your core OTUs are indeed shoot enriched in a quantitate way.

Reviewers' comments:

Reviewer's Responses to Questions

**Comments to the Author**

1. If the authors have adequately addressed your comments raised in a previous round of review and you feel that this manuscript is now acceptable for publication, you may indicate that here to bypass the “Comments to the Author” section, enter your conflict of interest statement in the “Confidential to Editor” section, and submit your "Accept" recommendation.

Reviewer #1: (No Response)

2. Is the manuscript technically sound, and do the data support the conclusions?

Reviewer #1: Partly

3. Has the statistical analysis been performed appropriately and rigorously? 

Reviewer #1: I Don't Know

4. Have the authors made all data underlying the findings in their manuscript fully available?

Reviewer #1: Yes

5. Is the manuscript presented in an intelligible fashion and written in standard English?

Reviewer #1: Yes

6. Review Comments to the Author

Reviewer #1: The authors have added a lot of text and edited their manuscript heavily - I am surprised to see a strong reluctance to remove the strong focus on DHA from the manuscript even though both reviewers are concerned about this. I would consider the manuscript suitable for publication without it.

They also added some additional introduction and discussion that I find unnecessary and rather irritating.

The conclusion is too long - part of it should be moved into the discussion i.e. Lines 616 - 630 should not be part of a conclusion

Looking into the use of just a few of references that were used here as mentioned below, I still think that the authors will have to do their due diligence and carefully check the context and content of all the references that they are using!

I hope that the authors are less reluctant to take advice this time. The data and biogeography aspect of this study is certainly worth publishing.

specific comments

Line 24 and 25. “association patterns congruent with those of a phyllosphere microbiome that mediates host physiology” this argument goes both ways - and can easily be explained by the plant host shaping the microbiome and not the microbiome shaping the host.

Line 52-64 neither me nor the other reviewer asked for this - and it is not appropriate in the context of a research paper - simply write which definition you use like you do any in line 46 - here you already write "the leaf surface, or phyllosphere" so this whole discussion is distracting from the study. I also cannot follow the whole point of this paragraph - and suggest to delete to whole paragraph from line 52-64.

additionally reference 10 does not support your statement. Whipps does not include the endophytic compartment in the phyllosphere - if you read carefully, they are writing about a surface area and distinguish phyllosphere bacteria from endophytes: Phyllosphere bacteria can promote plant growth and both suppress and stimulate the colonization and infection of tissues by plant pathogens (Lindow and Brandl 2003; Rasche et al. 2006a). Similarly, fungal endophytes of leaves may deter herbivores, protect against pathogens and increase drought tolerance (Arnold et al. 2003; Schweitzer et al. 2006).

Regarding reference 11 this review is really not great - first off, I agree that they falsely define the phyllosphere by including the endophytic compartment, they are very alone in this. They also proclaim highly problematic messages which were impressively dismissed just a year later it starts with "Most microorganisms of the phyllosphere are nonculturable in commonly used media and culture conditions" e.g. Bai et al. 2015 (nature) and many others showed that indeed most bacteria from leaf surfaces can be cultivated.

line 77 references 9,24 and 28 are not an appropriate reference here - these studies do not investigate interactions of the microbiota with the plant host instead the are investigating community structure but not micro-host interactions.

Line 128 - 130 this sentence still suggests that you would be investigating DHA variations - I only agree with the second part of the sentence that the manuka phyllosphere microbiome is underexplored. Reword the sentence to avoid overstressing DHA. Instead the authors may claim that the knowledge that they build here can in the future be useful to explore reasons for differing DHA concentrations.

Line 131 - 134 and again the authors are stressing that they will deliver answers to why DHA concentrations differ - your study does not contribute to extending the knowledge of this particular aspect and can only function as an hypothesis generator

General comment- there is a dichotomy between the core microbiome concept and the heterogeneity in DHA - which is still a prevailing problem of the underlying hypothesis: if the core microbiome were to be responsible for DHA formation, it should be more uniform. Instead, the accessory microbiome seems to have larger effects on the formation of DHA. Despite the claims of the authors that they adjusted their writing this is still reads like this is connected.

Line 148-150 Both reviewers asked you to remove this statement and this reviewer stands to this statement. Remove the claims that this increases the understanding about if microorganisms change DHA concentrations if DHA and or host physiology were not investigated.

Line 457-465 I reiterate - the PiCrust analysis does not add to the study and it is not discussed at all. It should be deleted. It also does not add worthwhile intel but rather highlights that in the future, metagenomic and transcriptomic studies should be conducted. Delete it

Line 505 regrading Terriglobus - now that is exciting - what is the known about this organism?

Line 518 - 535 With your study you show that a limited number of species are part of a core microbiome, you see this as evidence that they are important for community functioning and plant health but based on this data it is not possible if they are “key-stone” taxa (for further explanation about keystone taxa in the phyllosphere check e.g. the following papers https://journals.plos.org/plosbiology/article?id=10.1371/journal.pbio.1002352  https://pubmed.ncbi.nlm.nih.gov/31558832/). Since such an analysis is not possible and the results provided to not support the claims the authors make, I reject the response of the authors to point 4 of my previous review. No evidence was provide to support their claims.  

Line 530 diazotrophic nitrogen fixation is unlikely to be a major function on leaves in temperate conditions since it requires absence of oxygen reference 51 refers to tropical rain forest leaves which cannot be compared to temperate leaves since the leaves develop stratified biofilms that allow for anoxic niches. Reference 70 Delmotte et al did not investigate if the bacteria actually “provide various nutrients, phytohormones, and precursors for essential plant metabolites”. What we do know is that methylobacteria species and others metabolise methanol, some bacteria produce phytohormones (check studies regarding IAA of Leveau and others, or Rhodococcus that produce cytokine in studies from Danny Vereecke or Paula Jameson)

The discussion ends on a discussion around contamination this give a lot of emphasis on the used methodology however, previously asked to remove this for a reason and I will support this here once more: the claims that the authors make a merely based on assumptions and were not benchmarked against other established techniques. Instead I would move parts of this paragraph into the material and methods and indicate which educated decisions were made to change existing protocols.

7. PLOS authors have the option to publish the peer review history of their article (what does this mean?). If published, this will include your full peer review and any attached files.

Reviewer #1: No

---

## [Author Response · Author response to Decision Letter 1]

13 Jul 2020

Please see Response to Reviewers.pdf for our responses to specific reviewer and editor comments.

---

## [Decision Letter · Decision Letter 2]

21 Jul 2020

A core phyllosphere microbiome exists across distant populations of a tree species indigenous to New Zealand

PONE-D-20-02828R2

Dear Dr. Lee,

We’re pleased to inform you that your manuscript has been judged scientifically suitable for publication and will be formally accepted for publication once it meets all outstanding technical requirements.

Kind regards,

Omri Finkel, PhD

Academic Editor

PLOS ONE

Additional Editor Comments (optional):

Reviewers' comments:

Reviewer's Responses to Questions

**Comments to the Author**

1. If the authors have adequately addressed your comments raised in a previous round of review and you feel that this manuscript is now acceptable for publication, you may indicate that here to bypass the “Comments to the Author” section, enter your conflict of interest statement in the “Confidential to Editor” section, and submit your "Accept" recommendation.

Reviewer #1: All comments have been addressed

2. Is the manuscript technically sound, and do the data support the conclusions?

Reviewer #1: Yes

3. Has the statistical analysis been performed appropriately and rigorously? 

Reviewer #1: I Don't Know

4. Have the authors made all data underlying the findings in their manuscript fully available?

Reviewer #1: Yes

5. Is the manuscript presented in an intelligible fashion and written in standard English?

Reviewer #1: Yes

6. Review Comments to the Author

Reviewer #1: all my comments and requests were sufficiently addressed. The manuscripts presents a lot more concise and focuses on well-ground support by the authors findings and the literature.

7. PLOS authors have the option to publish the peer review history of their article (what does this mean?). If published, this will include your full peer review and any attached files.

Reviewer #1: No

---

## [Editor Report · Acceptance letter]

3 Aug 2020

PONE-D-20-02828R2 

A core phyllosphere microbiome exists across distant populations of a tree species indigenous to New Zealand 

Dear Dr. Lee:

I'm pleased to inform you that your manuscript has been deemed suitable for publication in PLOS ONE. Congratulations! Your manuscript is now with our production department. 

Kind regards, 

on behalf of

Dr. Omri Finkel 

Academic Editor

PLOS ONE